# Mechanism Understanding of Li-ion Separation Using A Perovskite-Based Membrane

**DOI:** 10.3390/membranes12111042

**Published:** 2022-10-26

**Authors:** Mahsa Golmohammadi, Meysam Habibi, Sima Rezvantalab, Yasin Mehdizadeh Chellehbari, Reza Maleki, Amir Razmjou

**Affiliations:** 1Department of Polymer Engineering & Color Technology, Amirkabir University of Technology, Tehran 15916-34311, Iran; 2School of Chemical Engineering, College of Engineering, University of Tehran, Tehran 14176-14411, Iran; 3Renewable Energies Department, Faculty of Chemical Engineering, Urmia University of Technology, Urmia 57166-419, Iran; 4Department of Chemical and Petroleum Engineering, Sharif University of Technology, Tehran 11155-9161, Iran; 5Computational Biology and Chemistry Group (CBCG), Universal Scientific Education and Research Network (USERN), Tehran 14197-33141, Iran; 6School of Engineering, Edith Cowan University, 270 Joondalup Drive, Joondalup, Perth 6027, WA, Australia; 7UNESCO Centre for Membrane Science and Technology, School of Chemical Engineering, University of New South Wales, Sydney 2052, NSW, Australia

**Keywords:** energy storage, lithium extraction, molecular dynamic simulation, ion transport, perovskite

## Abstract

Lithium ions play a crucial role in the energy storage industry. Finding suitable lithium-ion-conductive membranes is one of the important issues of energy storage studies. Hence, a perovskite-based membrane, Lithium Lanthanum Titanate (LLTO), was innovatively implemented in the presence and absence of solvents to precisely understand the mechanism of lithium ion separation. The ion-selective membrane’s mechanism and the perovskite-based membrane’s efficiency were investigated using Molecular Dynamic (MD) simulation. The results specified that the change in the ambient condition, pH, and temperature led to a shift in LLTO pore sizes. Based on the results, pH plays an undeniable role in facilitating lithium ion transmission through the membrane. It is noticeable that the hydrogen bond interaction between the ions and membrane led to an expanding pore size, from (1.07 Å) to (1.18–1.20 Å), successfully enriching lithium from seawater. However, this value in the absence of the solvent would have been 1.1 Å at 50 °C. It was found that increasing the temperature slightly impacted lithium extraction. The charge analysis exhibited that the trapping energies applied by the membrane to the first three ions (Li^+^, K^+^, and Na^+^) were more than the ions’ hydration energies. Therefore, Li^+^, K^+^, and Na^+^ were fully dehydrated, whereas Mg^2+^ was partially dehydrated and could not pass through the membrane. Evaluating the membrane window diameter, and the combined effect of the three key parameters (barrier energy, hydration energy, and binding energy) illustrates that the required energy to transport Li ions through the membrane is higher than that for other monovalent cations.

## 1. Introduction

Nowadays, Li is known as one of the most demanding materials with ever-increasing applications in different fields of research such as medicine [1,2], energy storage devices [3,4,5], air treatment products [6], Li-ion batteries [3,7,8], ceramics and glass [9,10,11], lubricating greases, polymer production, and alloys [12,13,14]. Li is the lightest alkali metal with a density of 0.534 g/cm^3^, which is electrochemically active with a high electrode potential of −3.05 V and has the highest specific heat capacity of any solid element. These properties make Li compounds very appealing in a variety of commercial applications. Based on the abovementioned applications, massive sources of Li compounds are expected to exist. Li resources are divided into two main categories: solids (e.g., mineral ores, recycled-waste Li-ion batteries, and electronic waste) and liquids (e.g., salt-lake brine, geothermal brine, and seawater). Seawater has the highest amount of Li resources (2.5 × 10^11^ tone), but the average Li concentration is low at 0.17 mg/L, which means it has a concentration of 0.1–0.2 ppm; thus, it is not of technical relevance in the spotlight. However, Li sources—deposits that contain Li [15]—are estimated to be equal to 21,000,000 metric tons [16], and will be exhausted by 2080 [17,18]. The extraction of Li can be performed via different methods such as precipitation [19,20], calcination methods [21], adsorption [22], electro-dialysis [23,24,25], electrolysis [26,27], and nanofiltration membranes [28,29,30,31,32,33]. Because of its simplicity and cost effectiveness, the adsorption process is a potential technology for Li recovery from seawater and salt-lake brine. Inorganic Li ions sieve with a high selectivity, capacity, and stability, and have obtained a lot of interest.

Among the aforementioned methods, Li extraction in the nanofiltration membranes has been conducted via a combination of the two main extraction mechanisms, including size sieving and charge repulsion effects [34]. These nanoporous membranes attract massive attention due to their desirable advantages, including their scale-up possibility [18,35], low energy consumption, facile operation, and environmental friendliness [36]. However, the decisive factor which makes nanofiltration membranes suitable for Li extraction is the adaptability of the membrane method with ocean solutions, which contain high concentrations of competing ions (e.g., Na^+^, Mg^2+^, Ca^2+^, K^+^, and Li^+^) [18]. The low Li^+^ concentration in solution but the high concentration of the interfering ions (e.g., Mg^2+^, Ca^2+^, Na^+^, and K^+^) is a significant obstacle for the highly efficient and environmentally friendly extraction of Li. In the nanofiltration membrane, Li permeation is usually measured by the extraction ratio of Li^+^ and Mg^2+^, which is known as Li^+^/Mg^2+^ selectivity [34]. For porous membranes, membrane porosity determines permeability, whereas membrane pore size determines selectivity. High permeability and selectivity require both high porosity and proper pore sizes. For instance, in some discussions, molecular dynamics simulations revealed that Li^+^ mobility was enhanced, whereas the K^+^ and Na^+^ mobility was reduced. Consequently, a higher selectivity of Li^+^ with K^+^ and Na^+^ was obtained. Meanwhile, for membrane selection, a better understanding of the trade-off between membrane permeability and Li selectivity is expected [37]. Therefore, it is essential to design Li-Selective Membranes (LSMs) to tackle the challenges in the path of Li extraction. In this regard, different porous structures were proposed and investigated computationally and experimentally for utilizing Li-selective membranes. For instance, Wang et al. studied the Li extraction efficiency of the A-site La_0.33_NbO_3_ perovskite structure. Based on their results, per La_0.33_NbO_3_ structure, near 0.67 Li were inserted electrochemically. Additionally, the charge transfer resistance in the La_0.33_NbO_3_ structure has an inverse relationship with the diffusion coefficient due to the Li insertion into the La_0.33_NbO_3_ perovskite structure [38]. In terms of computational investigation, the Li conductivity of Li_3_OCl_0.75_Br_0.25_ was predicted using an Ab Initio Molecular Dynamics (AIMD) simulation [39]. However, among the proposed structures, Lithium Lanthanum Titanate (LLTO) perovskite, Li_3x_La_(2/3−x)(1/3−2x)_TiO_3_ (0.04 < x < 0.16), attracts huge attention [40]. The plural A-site vacancies in its structure make it suitable for the seperation of the Li ions. Additionally, the bulk and grain-boundary conductivities of the LLTO structure at room temperature equal 10^−3^ S/cm and 10^−4^–10^−5^ S/cm, respectively [41], where the grain boundaries influence the Li^+^ diffusion through the LLTO samples. In the LLTO structure, a substantial number of “A” site vacancies are created by the disordered arrangement of Li and La ions, which facilitates Li ion transportation. To migrate from one “A” site to the next available site, Li^+^ ions have to pass through a bottleneck of four surrounding oxygen atoms [42]. Ionic conduction is caused by Li ion migration [43]. Many attempts have been made and discussed to increase ionic conductivity in the literature. Ionic conductivity (σ), an important property of bulk materials, defined as the velocity multiplied by the number of ions that are sequentially moving, is linked to Li migration on a micrometer length scale [44]. The Li diffusion pathways in the LLTO structure have been proposed, and the vacancies at the La site are important for Li^+^ migration [45]. High Li^+^ conductivity and selectivity in the extraction membranes are directly related to the crystal structure; in this respect, LLTO is known as one of the solid-state Li-ion superconductors [18]. In terms of the crystal structure, the LLTO structure contains TiO_6_ octahedra, which are interconnected to form cubic cages that can be suitable places for accommodating the Li^+^ and La^3+^ ions. The diameter of the created cages, which the four neighboring TiO_6_ tetrahedra have created, is 1.07 Å, and Li should be passed through them. However, by comparing the LLTO window diameter (1.07 Å) with the Li ion size (1.18 Å), it is obvious that the LLTO pore diameter is larger. Because of this miniature difference, small distortion and enlargement in the windows should happen [18].

In the current research, Molecular Dynamic (MD) simulations were conducted on the mechanism of Li extraction via the LLTO membrane to study the mechanism of the change in LLTO pores as well as the effect of the most dominant parameters (e.g., pH and temperature) on the LLTO pores expansion and Li extraction through them. If it is preferred to analyze the factors that could be used to distinguish Li^+^ ions from chemically identical monovalent and divalent ions in seawater, theoretical models and experimental verification reveal that the spacing, surface charge, functional group, and nanopore morphology are all significant for Li^+^ selectivity [46,47]. For this fact, in Li extraction from saline brines, the absolute separation efficiency depends on the relative selectivity of Li and other ions. The selective separation of Li^+^ and other ions requires molecular scale control of chemical/physical homogeneous nanomaterials. The lack of practical analytical methods within the nanoscale spatial and temporal resolution makes it difficult to fully comprehend the Li^+^ ion separation mechanism and pore geometry/chemistry interaction with Li^+^ ions. Although molecular simulations can provide insight into solute transport, the simulation capacity and membrane chemical heterogeneity preclude a complete depiction of ion transport through a realistic long pore. The results confirmed the undeniable role of pH and temperature on the LLTO pores size increment and facilitation of Li ions transmission through the membrane. Additionally, differences in hydration are important in explaining the extremely precise selectivity in cations. In the case of partial or complete dehydration during the process of transport, a more detailed approach is required to describe the interaction of water molecules with the ion as well as the interaction between the hydrated ion and the pore. The pore size directly influences ion transport, and this mechanism states that the energy barrier for ion transfer through small pores increases; hence, if water remains in these pores during the equilibrium, ion transport is the most hindered [46,47].

## 2. Simulation Methods

### 2.1. Study Plan

A simulation of the LLTO membrane with specific applications of Li extraction was performed in five different simulations, as outlined in Table 1.

**Simulation 1, 2, and 3**: Boxes in simulations 1 and 2 contain two LLTO structures and one Li ion. While in simulation 3, two LLTO membranes were placed in the middle of the simulation box surrounded by 500 ions (125 Li^+^, 125 K^+^, 125 Na^+^, and 125 Mg^2+^). In these three simulations, we aimed to investigate the effect of temperature and pH on the Li hydration, LLTO selectivity, and mechanisms of the LLTO pore changes. Finally, the interaction energies between the ions and membranes were scrutinized in more detail.

**Simulation 4 and 5:** In these cases, neither water nor other ions entered the simulation box; however, there were only two LLTO membranes in the vacuum condition. The objective of comparing the LLTO structure without water was to observe the impact of temperature on the expansion of membrane cavities. Moreover, the obtained results were also explained by utilizing molecular and atomic analyses, including interaction energies, hydration radii, etc.

### 2.2. Material and Methods

For the LLTO design, the LLTO’s CIF file was obtained from the Crystallography Open Database. Then, water molecules and ions were inserted into the simulation boxes with different concentrations and numbers using GROMACS software. The OPLSA force field and the tip3p water model were used to simulate water in these systems. The resulting structure was parameterized using the Polypargen or OBGMX sites and the GROMACS x2top command. Atom charges were calculated using CP2K software via ESP force fields in the Gaussian approach. The simulation box and LLTO structure were designed using the gmx editconf command and their sizes were equaled to 15 × 6.7 × 6.7 and 1.2 × 6.5 × 6.5 nm^3^, respectively.

The migration of the ions from the ionic part to the ion-free part in this system is caused by the concentration gradients, which act as a driving force; thus, the ions pass through the LLTO membrane. When the concentration changes, the bulk molecules suffer changes caused by the competition of three main forces: Coulomb interactions, due to the presence of an electrical charge; a steric interactions force, produced by the overlap of the molecules; and the elastic forces, due to entropic effects. These forces will be discussed in detail in the simulation 2 explanation. Considering that the membrane is a perovskite, it can suffer expansion according to its physicochemical properties. When increasing the concentration, an osmotic pressure appears and tries to separate them (steric interaction), and as a consequence, an elastic force appears because of the change in the conformational entropy. The electric repulsion produces an expansion of the pore size and also contributes to the overlapping between molecules. Under this behavior, the steric interaction and the elastic force control the pore size.

The simulation times were also considered 100 ns. The LINCS algorithm was also used to constrain the hydrogen bonds. In NVT and NPT simulations, Berendsen thermostat and barostat algorithms were used to stabilize the temperature at adjusted temperatures and pressure at 1 bar, respectively, with a simulation time of 600 ps. Additionally, the Nose–Hoover thermostat and Parrinello–Rahman barostat were applied in the MD simulation to stabilize the temperature and pressure during the 100 ns simulation, respectively. The simulation box was first minimized with a steep integrator. Then, it was equilibrated with the algorithms for 0.5 nanoseconds in the NVT and NPT ensembles. Finally, the simulations were performed for 100 ns. Coulomb energy relations and Lennard-Jones energy were used to calculate electrostatic energy and van der Waals (vdW) energy. The basis for calculating their energy was based on the average distance between the charges of the atoms and the geometric mean distance of the charges of the atoms from each other, respectively. The electric charge corresponds to each atom. K is also a constant of the Coulomb energy equation [48,49]. The average energy at each time was also obtained from the simulation. All the hydrogen-containing bonds were restrained using the LINCS algorithm [50].

## 3. Results and Discussions

The elucidation of ion transport through membranes is an important issue to improve membrane separation. Steric interaction, electrostatic, dielectric, vdW, frictional, and viscous interactions are the various mechanisms of transporting ions through the membranes. The steric effect poses a high energy barrier, particularly for rigid pores with dimensions smaller than the bare size of the ion. Electrostatic interactions and the intrinsic charges at the pore entrance are further affected by selectivity, which favors counter-ions diffusion but excludes co-ions. For a solute with a hydration diameter (d_h_) larger than the pore size (d_p_), the necessary dehydration (or partial dehydration) sets an energy barrier as well [46]. A dehydration reaction is a chemical reaction between two compounds where one of them is water that involves the loss of water from the reacting molecule or ion. The dielectric effect regulates the energy barrier during ion permeability once the ions become partially dehydrated and penetrate the pore, resulting in the selectivity of similar monovalent ions across membranes. After penetrating the pore, pore surface friction and viscous drag forces play an essential role in elucidating its mechanism, so that the affinity of the pore wall with the electrostatic and vdW forces affects solute hindering [46]. Designing a highly selective membrane requires a rational incorporation of the above mechanisms.

The current study addresses two aspects of ion transport through the LLTO membrane: the impact of pH and temperature in two main parts. The first part comprises three individual simulations to investigate the pH of the solution. Figure 1a–c depict visual highlights observed in the transportation of the single solvated Li^+^ ion through the LLTO membrane at pH 7 and 5.5.

Based on the results of the first simulation, we observed no difference in the LLTO membrane at room temperature and pH 7. Apart from the electrical interactions, the Grotthuss-like mechanism [51] describes the separation mechanism for the neutral media. The ions are conducted via the two-surface charge-governed transport mechanism and Grotthuss-like mechanism by jumping from one membrane side to another while moving in the direction of the electric field. Additionally, mass transfer is carried from the bulk to the boundary before transporting inside the membrane. Combining this theory with an appropriate concentration driving force model can determine the performance of membrane and selectivity. Based on the results, the membrane’s pore size confronted no variation and remained at 1.07 Å during the simulation (Figure 1b).

In the second simulation, the pH of the water solution was altered to 5.5 and the temperature of the simulation box was adjusted to 50 °C. Figure 1c displays the dehydration and permeation of Li^+^ ions through the LLTO membrane, which has expanded pores in the acidic pH. The snapshot shows the peeling of the water shell off the Li^+^ ion while passing through the enlarged pore. As mentioned earlier, the pore size of the LLTO membrane at pH 5.5 expanded to 1.18–1.2 Å. The miniature expansion (0.03–0.13 Å) in the size of the LLTO pores can be attributed to the change in both the temperature and pH value. The pore size is a function of the temperature and hydrogen bond interaction between ions and the membrane.

In the second simulation, we sought to analyze the impact of charge (acidic condition). As is well known, Li^+^ cations attract oxygen ions in the solutions while repelling hydrogen cations because they have the same charge. The lithium ions were dehydrated completely due to the simultaneous existence of repulsive and attractive forces between lithium ions and the LLTO membrane. Figure 1d shows that the interaction energies of Li^+^-H^+^ was repulsive, while the Li^+^-O^2−^ interaction energies are shown in Figure 1e as being attractive forces. These simultaneous repulsive–attractive forces pull the competing ions back and forth, which can be deemed one of the leading reasons for the ions’ dehydration. In addition to the lithium ion dehydration, these repulsive–attractive interactions caused the LLTO membrane’s pore size to expand up to 1.18 to 1.2 Å, which can bring efficient lithium selectivity for the LLTO membrane.

We investigated the interaction of Li^+^ with the LLTO membrane after analyzing the interactions of O^2−^ and H^+^ ions with Li^+^. In this regard, the surface charge of the LLTO membrane was calculated (Figure 1f); because of the negative charge of the LLTO membrane, positive lithium ions were attracted to it and were kept as close to the membrane as possible. The Li ions were thus affected by the forces of attraction from the LLTO membrane side. The Lithium ions moved closer to the membrane during this period and became dehydrated when positioned between the attraction–repulsion forces. As a result of the lithium ion dehydration, the possibility of lithium transferring across the membrane will be provided. Notably, increased temperature and acidic conditions expand the LLTO membrane’s pore size and make it easier for lithium ions to pass through the LLTO membrane.

In order to conduct further research, average hydrogen bond interactions between Li^+^ and the LLTO membrane along with the three simulations were quantified (Figure 2a) [52]. As can be seen, among the simulations 1, 2, and 3, the number of hydrogen bonds in the latter simulation increased the H bonds to 54 during the simulation, which can be attributed to both the pH and the existence of other ions as well as lithium in the simulation box. The slight difference in the number of hydrogen bonds between the second simulation (average H bond = 49) and third simulation can be related to the number of competing ions. The solution pH also plays a critical role in local and long-range ion−water interactions. It is thus reasonable to expect the solution pH to influence the structure and size of solvated ions. Hydrogen bonding interaction changed membrane pore parameters and covered the membrane surface. The presence of more ions in a water solution causes the formation of more hydrogen bonds in the system, increasing the total number of hydrogen bonds.

After filtration, the lithium ions formed almost no monohydrate (H_2_O) Li^+^. The low proportion of lithium monohydrates may be attributed to the abnormally low proportion of monohydrates in the bulk solution because of the high charge density of Li^+^ and high hydration energy. Regardless of the differences between these alkali metal ions, dehydration appears to be ubiquitous when the pore size is proportional to the hydrated size of the ion.

To provide deeper insight into the governed interaction energies, the first part of the simulations was compared, as seen in Figure 2b. The relative similarity in interaction energies for simulations 2 and 3 can be attributed to their similar conditions (50 °C and pH 5.5); however, due to the existence of more ions in the third simulation, it possesses a slightly higher difference in interaction energies compared with the second simulation. By comparing simulations 1 and 2, it can be firmly confirmed that the stronger interactions in simulation 2 were due to the higher temperature and acidic condition. Subsequently, the results approve the conditioned high Li^+^ selectivity in the third simulation. In other words, acidic conditions and higher temperatures provide a better situation to achieve higher Li selectivity over other existing ions (Figure 2c).

In the third simulation, two LLTO membrane structures were surrounded by 500 ions (125 Li^+^, 125 K^+^, 125 Na^+^, and 125 Mg^2+^) in the solution condition of pH 5.5 at 50 °C. Figure 2d reveals the effect of the presence of other ions in the separation of Li^+^ ions to visually represent the ratio of ions that passed the barrier.

The interaction energies between the Li^+^ and LLTO membranes in the presence of other cations (simulation 3) were investigated (Figure 2e). The negative sign of the interaction energies approves the existence of the attractive forces among the Li^+^, Na^+^, K^+^, and Mg^2+^ ions and the LLTO structure. As can be noticed, the vdW interactions had relatively minor contributions to the total energies, and its negligible value stemmed from the small radii of the competing ions. Thus, the constituent part of the total energies was formed by the electrostatic interaction energies, consisting of the ion–ion, water–ion, and water–water interactions energies. Electrostatic energies are more determinative in the membrane selectivity, in which more differences in the electrostatic interaction energies lead to higher selectivity and better membrane performance. The total energy of Li^+^, Na^+^, K^+^, and Mg^2+^ were −790 kJ/mol, −850 kJ/mol, −940 kJ/mol, and −1040 kJ/mol, respectively. By comparing the ion’s total interaction energies, it is obvious that they were higher than the hydration energies of Li^+^, Na^+^, K^+^, and Mg^2+^, which are equal to −515, −405, −321, and −1922 kJ/mol [53]. Suppose the trapping energy or total energy is stronger than the hydration energy of each ion. In that case, they can overcome the ion’s hydration energies, and as a consequence, the ions will be dehydrated. The obtained results showed that only Na^+^, K^+^, and Li^+^ could be fully dehydrated among present ions, and Mg^2+^ ions were partially dehydrated.

The fully dehydrated diameter of Li^+^ (0.136 nm) was only slightly larger than that of Mg^2+^ (0.13 nm), but was comparably smaller than that of Na^+^ (0.19 nm) and K^+^ (0.266 nm). The Na^+^ and K^+^ dehydrated diameters were larger than the LLTO membrane pore size, and the Mg^2+^ ions were partially dehydrated and consequently could not pass through the membrane (Figure 3). Li^+^ had a whole situation to pass through the membrane. The results exhibited the remarkable Li selectivity of the LLTO membrane against other ions. Nanofiltration under the various conditions indicated that dehydration was not merely driven by the complete exclusion of larger Mg^2+^ hydrates from the membrane; rather, a partial dehydration-driven transformation was occurring. In other words, the large Mg^2+^ hydrates were reducing their effective size by shedding part of their solvation shell to partition into and permeate across the nanometer membrane pores. In this case, the synergic effect of the temperature increment and pH decrement can help the selectivity of the LLTO membrane by widening its pores and effective ion dehydration. The difficulties in separating Mg^2+^ from Li^+^ lie in their similar ionic radii, either in bare or hydration states (Table 2). The fully dehydrated radius of Li^+^ (0.068 nm) was only slightly larger than that of Mg^2+^ (0.065 nm), but the hydration energy of Mg^2+^ was comparably larger than the trapping energy. Therefore, the Mg^2+^ ions could not pass through the LLTO membrane. Additionally, the hydration number of Li^+^ is five. Still, the lifetime or exchange rate of the water molecules in the hydration layer was 5 × 10^−9^ s, while Mg^2+^ had a hydration number of six with a significantly much slower exchange rate of 10^−6^ s. Because the extraction process took place at the aqueous/organic interface, the ions needed to overcome an energy barrier to cross the interface [46].

In the fourth and fifth simulations, the LLTO structure was placed in the simulation box without any ions or solvent. In these cases, only temperature changes from 25 °C to 50 °C occurred in simulations 4 and 5, respectively. As it is well known, temperature is an important factor governing mass transfer in membrane separation processes and has a significant impact on NF membrane performance. By comparing the results of these two simulations, the slight effect of the temperature change on the pore expansion was firmly confirmed (Figure 3). At 25 °C, the pore size remained at 1.07 nm (Figure 3a), while at 50 °C, the pore size in the LLTO membrane increased to 1.1 Å (Figure 3b). It can be seen that the pore size changed depending on temperature.

## 4. Conclusions

In the case of ion separation, developing membranes with nanopores tailored to match the ion solvation structure could significantly enhance ion−ion selectivity and lead to increased energy and process efficiencies. The novelty in this study lies in systematically evaluating the separation and dehydration mechanism based on the desired driving force during pore transport for small ions with Molecular Dynamics (MD). The LLTO membrane, which was employed as a high-performance membrane in the Li extraction application, was simulated in different environmental conditions (e.g., temperature and pH) via molecular dynamics. The results of this research indicated that three main factors have a significant positive effect on the performance of the Li extraction of LLTO membranes, including acidic pH conditions; repulsion–attraction forces, which led to the ions’ dehydration; and finally the efficient affiliation between the LLTO structure and Li ions, which led to a higher rate of Li permission to the LLTO membrane due to the pore size expansion from 1.07 Å to approx. 1.2 Å. Our simulations therefore suggest that ion dehydration is important to transport when the size of the nanopore is smaller than the hydrated ion, leading to water molecule shedding from the outermost solvation layer. Filtration by LLTO membranes induced ion dehydration for all the various alkali metal ions (Li^+^, Na^+^, and K^+^) in the bulk solution investigated. The shedding of the water molecules from the solvation shell allowed ions to enter pores that were smaller than their bulk hydrated size. With the current MD simulation, the dehydration of ions was found to mainly dominate the ion selectivity of the pores, and the difficulty of the ion dehydration could be reflected by the ion energy barrier. According to the results, monovalent cations can pass more easily through the nanopores than the divalent cations due to their smaller energy barriers, indicating that the pores are more effective in separating these two kinds of cations. As a final point, three important conclusions should be mentioned: Firstly, energetic barriers were strongly dependent on pore size. Additionally, there were distinct regimes related to the required dehydration. Transport was strongly hindered when the size of the pore was smaller than the hydrated radius. Second, energy barriers were determined by ion type (and thus hydration properties), and ion selectivity varied depending on pore size. Finally, and perhaps most crucially, dehydration was the primary method of ion transfer in the pores. Even when charge repulsion was taken into account, partial dehydration was the dominant predictor of the energy barriers for small, strongly hydrated ions whose hydrated radius was bigger than the pore size.

## Figures and Tables

**Figure 1 membranes-12-01042-f001:**
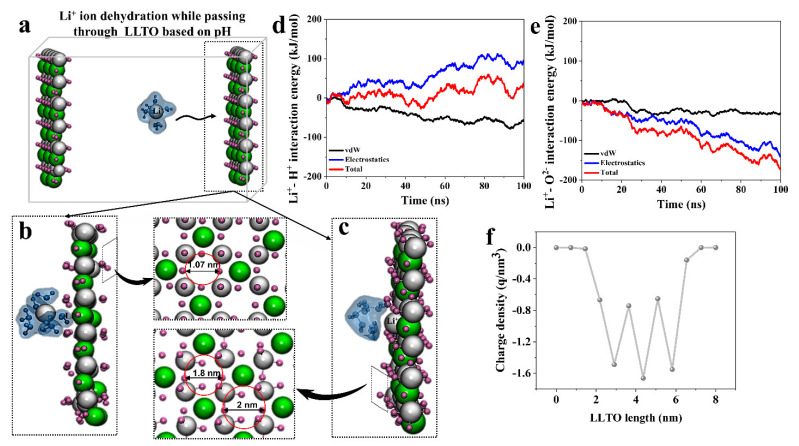
Ion transportation through the LLTO membrane: (**a**) initial conformation of solvated Li^+^ ion in the simulation box containing two LLTO membranes; (**b**,**c**) dehydration and permeation of Li^+^ ion at pH 7 and 5.5, respectively. Additional insets provide details on the effect of pH and temperature on the pore size; (**d**,**e**) energy analysis of Li^+^-H^+^ and Li^+^-O^2−^ interactions with the LLTO membrane, respectively; and (**f**) LLTO membrane’s surface charge.

**Figure 2 membranes-12-01042-f002:**
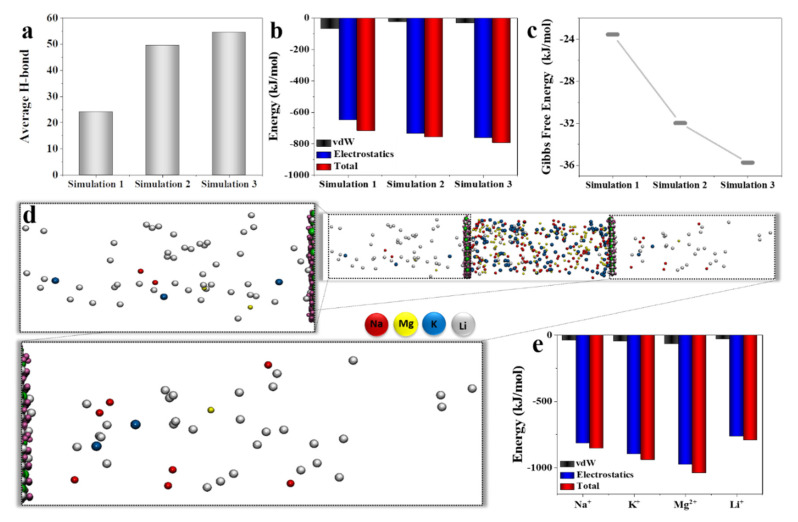
Li transportation through LLTO membrane in the presence of other cations: (**a**) average H bond between Li^+^ ion and LLTO membrane for the first part of simulations; (**b**) interaction energy between Li^+^ ion and LLTO membrane for the first part of simulations. Obviously, Li^+^ ion has the strongest interactions with the LLTO membrane in the 3rd simulation due to the pH and temperature conditions; (**c**) Gibbs free energy for three simulations reveals that the 3rd simulation displays the most stable interaction between Li^+^ ion and LLTO membrane; (**d**) transportation of ions through the membranes in the presence of 500 ions. Closer snapshots of both sides clearly show the high selectivity of the membrane toward Li^+^ ion in comparison with other ions (K, Mg, and Na); and (**e**) interaction energy of cations with the LLTO membrane in the 3rd simulation.

**Figure 3 membranes-12-01042-f003:**
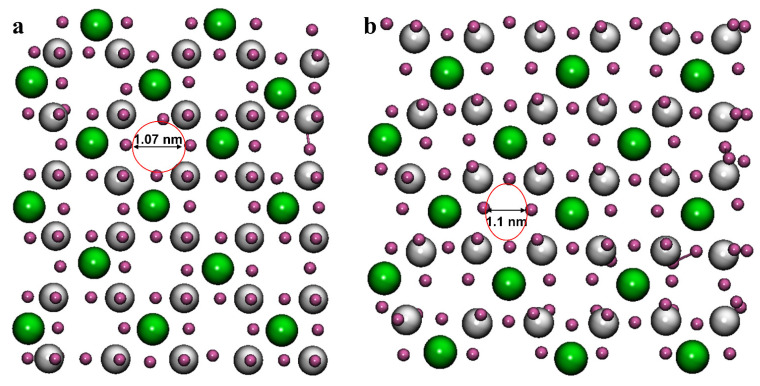
The impact of temperature on the pore size of LLTO membrane: (**a**) T = 298 K, (**b**) T = 323 K.

**Table 1 membranes-12-01042-t001:** Characteristics of all simulations.

Part	Simulation No.	Temp. (K)	pH	Solvent	Num. Of Ions/Ion Type
1st	1	298	7	Water	1 Li^+^
	2	323	5.5	Water	1 Li^+^
	3	323	5.5	Water	125 Li^+^, 125 K^+^, 125 Na^+^, and 125 Mg^2+^
2nd	4	298	-	-	0
	5	323	-	-	0

**Table 2 membranes-12-01042-t002:** Characteristics of the common cations and anions in the salt-lake brine.

Ion	Bare Ion Radius	Hydrated Radius	Hydration Number	Diffusion Coefficient	Hydration Free Energy
	[nm]	[nm]	±1	(10^−9^ m^2^ s^−1^ )	[kJ. mol^−1^]
H_3_O^+^	-	0.28	3	9.31	-
Li^+^	0.068	0.38	5	1.03	474
Na^+^	0.095	0.36	4	1.33	364
K^+^	0.133	0.33	3	1.96	295
Mg^+^	0.065	0.43	6	0.71	1828

## Data Availability

Not applicable.

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
