# Peer review of "Mechanism Understanding of Li-ion Separation Using A Perovskite-Based Membrane"

_membranes, 2022, doi:10.3390/membranes12111042_

Round 1

Reviewer 1 Report

The work under review can be accepted for publication in its current form.

Author Response

Many thanks to Referee 1 for your recommendation about our article. It is an honor for us.

Reviewer 2 Report

In this article that the mechanism of the ion-selective membrane and efficiency of the perovskite-based membrane have been investigated using molecular dynamic (MD) simulation. The results specified that the change in the ambient condition, pH, and temperature leads to the shift in LLTO pore sizes. The hydrogen bond interaction between ions and the membrane leads to expanding pore size, to facilitate lithium ions transmission through the membrane for enriching lithium from seawater, but increasing temperature slightly impacts lithium extraction. The study might be of some interest for the readers of Membranes. Although some new results reported in the manuscript, but some ambiguities should be well addressed prior to a potential publication in the journal. 

(1)The algorithm used in MD simulations in the paper needed further explanation.

(2)How to verify whether the dehydration of Li+ ion through LLTO membrane by the experiments and measurements?

(3)According to the simulation results in Figure 1, the interaction energies of Li+-H+ and Li+-O2- could change with time and their theoretical basis needed to be further supplemented.

(4)It needs to be supplemented to explain how to verify the results obtained by different simulations, such as the membrane’s pore size, the number of hydrogen bonds, and the interaction energy between ion and membrane, etc.

(5)Based on the hydration number of 5, How did the author conclude that the average H-bond=49 between Li + and LLTO membrane? That is, how to verify whether the results obtained by the 2nd simulation are correct. 

Author Response

Many thanks to Referee 2 for your detailed and watchful review of the submission. I've addressed the issues and your precise comments as follows:

  1. Your comment about The algorithm used in MD simulations in the paper needed further explanation.”

We appreciate your comment. As your recommendation, we tried to add more information about the details of our MD simulations as mentioned in the manuscript:

Coulomb energy relations and Lenard-jones energy are used to calculate electrostatic energy and van der Waals (vdW) energy. The basis for calculating their energy is based on the average distance between the charges of the atoms and the geometric mean distance of the charges of the atoms from each other, respectively. The electric charge corresponds to each atom. K is also a constant of the Coulomb energy equation [49,50]. The average energy at each time is obtained from the simulation, too. All the hydrogen containing bonds were restrained using the LINCS algorithm [51].

  1. Your comment about How to verify whether the dehydration of Li+ ion through LLTO membrane by the experiments and measurements?

Thanks for your precise comment. As you mentioned, the validation of the MD simulation is a principal task, which should be carefully addressed. But, as you know absolute relying on the experimental data for validating the results of the MD simulation could be challenging because experimental and computational approaches (simulation) are not the same in the space and time scale (https://pubs.acs.org/doi/10.1021/acs.jpcb.8b02144).

In this regard, we perform our simulation in different conditions, which let us compare LLTO pore size expansion and lithium-ion diameters in various conditions in the terms of temperature and pH. As mentioned in our manuscript, we perform the first simulation in a neutral condition (298 K and pH 7) and the pore size of the LLTO membrane did not show any variation and remained at 1.07 Å from the start till the end of the simulation. However, via comparing the results of the 2nd and 3th simulations, it is demonstrated that pH 5.5 and tuning the temperature at 323 K, can provide desirable conditions in which hydrated lithium ions are dehydrated and as a result, their ionic diameter will be reduced as well as they are able to pass through the pores of the LLTO membranes.

  1. Your comment about According to the simulation results in Figure 1, the interaction energies of Li+-H+ and Li+-O2- could change with time and their theoretical basis needed to be further supplemented.”

We appreciate the reviewer’s accurate evaluation. As mentioned in our manuscript, we just investigate the mechanism of the LLTO pores expansion under dominant parameters including pH, and temperature. Due to avoid making the simulation more complex, we didn’t consider time as a new parameter, in our 5 performed simulations. All simulations are performed at specific times, as mentioned in our article: “In NVT and NPT simulations, Berendsen thermostat and barostat algorithms were used to stabilize the temperature at adjusted temperatures and pressure at 1 bar, respectively with the simulation time of 600 ps. Also, in the MD simulation, the Nose–Hoover thermostat and Parrinello-Rahman barostat were applied for stabilization of the temperature and pressure during the 100 ns simulation, respectively. The simulation box was first minimized with an integrator, steep. Then, it was equilibrated with the algorithms for 0.5 nanoseconds in the NVT and NPT ensembles. Finally, the simulations were performed for 100 ns.”

Besides, it is reported (https://pubs.acs.org/doi/10.1021/acs.jpcb.8b02144) that multiple short simulations result in a better understanding of the simulated problem and its conditions rather than performing a single long-term simulation.

However, we are working on investigating other aspects and effective parameters on the separation performance of the LLTO membranes that their results will be included in our next publication, which will be published in the near future.

  1. Your comment about It needs to be supplemented to explain how to verify the results obtained by different simulations, such as the membrane’s pore size, the number of hydrogen bonds, and the interaction energy between ion and membrane, etc”

Thank you for your valuable comment. As you know absolute relying on the experimental data for validating the results of the MD simulation could be challenging, because experimental and computational approaches (simulation) are not the same in the space and time scale (https://pubs.acs.org/doi/10.1021/acs.jpcb.8b02144).

However, as you mentioned too, taking a glance at the results of the experimental investigation and comparing them can be one of the suggested strategies to validate our simulation results. In this regard, our findings are compared with the following experimental articles.

  • Sun, P. Guan, Y. Liu, H. Xu, S. Li, D. Chu, Recent Progress in Lithium Lanthanum Titanate Electrolyte towards All Solid-State Lithium Ion Secondary Battery, Crit. Rev. Solid State Mater. Sci. 44 (2019) 265–282. https://doi.org/10.1080/10408436.2018.1485551.
  • Inaguma, J. Yu, Y. Shan, M. Itoh, T. Nakamuraa, The Effect of the Hydrostatic Pressure on the Ionic Conductivity in a Perovskite Lanthanum Lithium Titanate, J. Electrochem. Soc. 142 (1995) L8–L11. https://doi.org/10.1149/1.2043988/XML.
  • Li, C. Li, X. Liu, L. Cao, P. Li, R. Wei, X. Li, D. Guo, K.W. Huang, Z. Lai, Continuous electrical pumping membrane process for seawater lithium mining, Energy Environ. Sci. 14 (2021) 3152–3159. https://doi.org/10.1039/D1EE00354B.
  1. Your comment about “Based on the hydration number of 5, How did the author conclude that the average H-bond=49 between Li+ and LLTO membrane? That is, how to verify whether the results obtained by the 2nd simulation are correct.”

We appreciate the reviewer’s accurate evaluation. As mentioned in the revised version of our article, we used LINCS algorithm for analyzing the available hydrogen bonds between all possible donors and acceptors. More explanation about this algorithm can be found in (https://manual.gromacs.org/current/reference-manual/analysis/hydrogen-bonds.html)

As mentioned in our article:

All the hydrogen containing bonds were restrained using the LINCS algorithm [51].

Also, many articles used the LINCS algorithm to analyze H bonds in various fields of research:

  • Peng, S. Wang, X. Wang, J. Gao, D. Xu, L. Zhang, Y. Wang, Liquid-liquid extraction and mechanism exploration for separation of mixture 2,2,3,3-Tetrafluoro-1-propanol and water using pyridine-based ionic liquids, J. Mol. Liq. 360 (2022) 119468. https://doi.org/10.1016/J.MOLLIQ.2022.119468.
  • Biswas, B.S. Mallik, Ultrafast Aqueous Dynamics in Concentrated Electrolytic Solutions of Lithium Salt and Ionic Liquid, J. Phys. Chem. B. 124 (2020) 9898–9912. https://doi.org/10.1021/acs.jpcb.0c06221 .
  • A. Lochbaum, A.K. Chew, X. Zhang, V. Rotello, R.C. Van Lehn, J.A. Pedersen, Lipophilicity of Cationic Ligands Promotes Irreversible Adsorption of Nanoparticles to Lipid Bilayers, ACS Nano. 15 (2021) 6562–6572. https://doi.org/10.1021/acsnano.0c09732.

Reviewer 3 Report

The manuscript is suitable for pubblication

This is an interesting work where Authors report on the implementation of Molecular Dynamic (MD) simulations to investigate the mechanism of Li extraction via Lithium Lanthanum Titanate (LLTO) membranes under the effect of the most dominant parameters (e.g. pH and temperature) and on the LLTO pores expansion. Authors studied the separation and dehydration mechanism based on desired driving force during pore transport. Simulations highlighted the role of ion dehydration when the size of the nanopore is smaller than the hydrated ion, leading to water molecule shedding from the eternal solvation layer. Results indicated that three main factors have a significant positive effect on the performance of Li extraction of LLTO membrane: acidic pH conditions; repulsion-attraction forces which lead to the ions dehydration; the efficient affiliation between LLTO structure and Li ions that leads to a higher rate of Li permission to the LLTO membrane due to the pore size expansion. Since developing membranes with nanopores tailored to match the ion solvation structure could significantly enhance ion−ion selectivity and lead to increased energy and process efficiencies, this work is highly valuable and could be published in its current form.

Author Response

Many thanks to Referee 3 for your recommendation about publishing our article.